# Spatiotemporal Solar Visibility Modeling for Lunar Landing Site Evaluation

Junseo Moon    Seokju Lee[†]

Korea Institute of Energy Technology (KENTECH)

{jsmoon, slee}@kentech.ac.kr

## Abstract

*Lunar landing site evaluation benefits from considering both terrain geometry and time-varying solar illumination, particularly in polar regions where extended eclipses can affect power availability, thermal conditions, and surface operations. We construct a spatiotemporal solar visibility representation over lunar terrain using digital elevation models, DEM-based horizon analysis, and SPICE-derived solar geometry for one synodic cycle. From the resulting binary visibility sequences, we derive mean solar visibility, longest continuous illumination, and longest eclipse duration, and incorporate these descriptors into a self-organizing map (SOM) to generate ranking and suitability maps. We evaluate the method across four lunar south polar regions and at ten previously proposed and three mission-associated coordinates, including Chandrayaan-3 Statio Shiv Shakti, IM-1 near Malapert A, and IM-2 in the Mons Mouton region. Results show that sites with similar mean solar visibility can exhibit different illumination interval structures and suitability outcomes. The site-level comparison further shows that the temporally extended SOM modifies local rank assignments relative to the mean-visibility baseline, with most analyzed coordinates receiving lower, more favorable ranks after temporal descriptors are included. These results suggest that spatiotemporal solar visibility modeling provides information beyond mean solar visibility alone and can support more context-aware lunar south polar landing site evaluation.*

## 1. Introduction

Solar illumination is a primary environmental constraint in lunar surface exploration because it directly affects photovoltaic power availability, thermal control, radiative heating conditions, and mission operations for landers and rovers [4, 6, 8]. This constraint becomes particularly important in polar regions, where low solar elevation and terrain-induced horizon occlusion can produce long eclipses and

highly nonuniform illumination over short spatial scales [5, 7, 15]. As a result, lunar polar illumination is naturally posed as a terrain-aware and time-varying visibility problem, for which a spatiotemporal representation is more informative than a single averaged scalar.

The lunar south polar region is a particularly relevant setting for this problem. Because of the Moon's small axial tilt, nearby sites can exhibit substantially different temporal illumination patterns even when their mean solar visibility is similar [7, 15]. Prior illumination studies have therefore considered operationally relevant quantities beyond average illumination [5, 15]. Recent studies have also examined extended eclipse intervals and quasi-continuous illumination conditions in south polar analysis [23, 25]. These observations suggest that a time-averaged visibility measure alone may be insufficient for characterizing temporal risk in candidate landing regions.

Lunar landing site evaluation is increasingly treated as a multi-factor problem that must balance engineering constraints, environmental conditions, and scientific value. Recent studies have explored south polar landing site assessment using terrain, illumination, slope, and indicators derived from remote sensing [9, 13, 18]. In particular, Zhang *et al.* [24] proposed candidate soft-landing sites in several lunar south polar regions, including Shackleton, Haworth, Cabeus, and Shoemaker. In parallel, self-organizing maps (SOMs) provide an unsupervised framework for structure-preserving clustering and multi-feature evaluation [11, 22], and they have recently been applied to comprehensive lunar south polar landing site evaluation using aligned scientific and engineering feature layers [14]. However, in these evaluation frameworks, solar visibility is typically represented by a single time-averaged value, which does not explicitly preserve the temporal structure of illumination and eclipse intervals.

This limitation is relevant not only to regional suitability mapping but also to the interpretation of individual landing-site coordinates. Regional maps and aggregate suitable-area ratios can show broad spatial changes, but they do not directly indicate whether temporal illumination descriptors alter the rank assigned to specific coordinates. Consequently,

---

[†]Corresponding author.

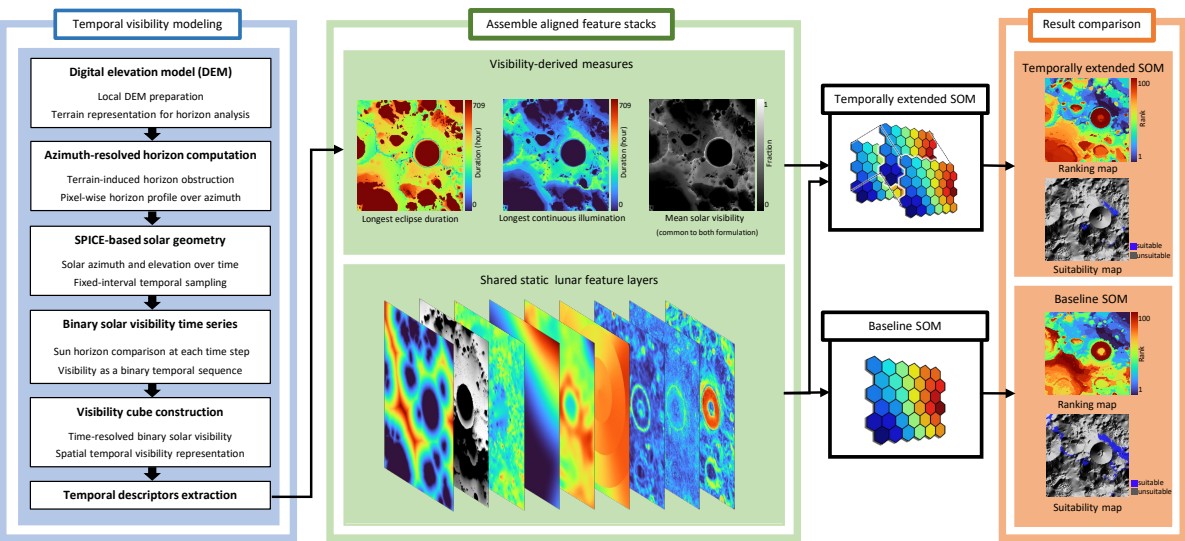

Figure 1. Overview of the proposed workflow. DEM-based horizon analysis and SPICE-derived solar geometry are used to convert local terrain into a time-resolved binary solar visibility cube, from which mean solar visibility, longest continuous illumination, and longest eclipse duration are derived. The baseline SOM follows the mean-visibility formulation used in prior SOM-based landing-site evaluation [14], while the temporally extended SOM additionally incorporates the two interval-based descriptors. Both formulations use the shared static lunar feature layers and produce ranking and thresholded suitability maps, with lower rank values indicating more favorable landing conditions.

the site-level effect of temporally extended evaluation remains unclear for coordinates associated with previous site proposals, lunar landings, or landing attempts.

In this work, we construct a spatiotemporal solar visibility representation over 3D terrain. The representation combines lunar digital elevation models [3, 12, 19], DEM-based horizon analysis [20], and SPICE-derived solar geometry [1, 2]. From the resulting visibility sequence, we derive temporal illumination descriptors, including mean solar visibility, longest continuous illumination, and longest eclipse duration, and examine their value in downstream lunar landing site evaluation using a SOM-based framework. The baseline formulation uses mean solar visibility together with shared static lunar feature layers, whereas the temporally extended formulation additionally incorporates the two interval-based descriptors.

We evaluate the proposed representation in four lunar south polar regions associated with the candidate sites of Zhang *et al.* [24]: Shackleton, Haworth, Cabeus, and Shoemaker. We compare the resulting ranking and suitability maps with those from a baseline formulation using mean solar visibility. To further examine site-level effects, we query the baseline and temporally extended SOM outputs at ten previously proposed coordinates and three mission-associated coordinates: Chandrayaan-3 Statio Shiv Shakti [21], IM-1 near Malapert A [16], and IM-2 in the Mons Mouton region [17]

Table 1. Comparison of the input representations used in the baseline SOM and the temporally extended SOM.

| Item | Baseline SOM | Temporally extended SOM |
|---|---|---|
| Common solar measure | Mean solar visibility | Mean solar visibility |
| Additional temporal descriptors | None | Longest continuous illumination, longest eclipse duration |
| Shared static inputs | Static lunar feature layers | Same as baseline |
| Temporal source | Time-resolved visibility summarized as mean solar visibility | Time-resolved binary visibility used to derive mean and interval descriptors |

## 2. Method

### 2.1. Temporal Solar Visibility Modeling

We model solar visibility as a terrain-constrained process that varies over time, rather than representing it only by its mean value. As illustrated in Figure 1, the workflow begins with a local digital elevation model, from which terrain horizon obstruction is estimated for each region of interest (ROI). This horizon information is then combined with SPICE-derived solar geometry to evaluate solar visibility over one synodic cycle.

For each pixel, horizon obstruction is computed as a function of azimuth so that terrain occlusion can be evaluated with respect to the instantaneous solar direction. Solar position is sampled at fixed time intervals, and solar visibility is defined in binary form at each time step. A value of 1 indicates that the Sun is visible above the local terrain horizon in the corresponding azimuthal direction, whereas a value of 0 indicates that it is occluded by surrounding topography. Repeating this evaluation over the full time grid yields a binary solar visibility sequence for each pixel, and stacking these sequences over the ROI forms a time-resolved visibility cube.

From this visibility sequence, we derive three visibility descriptors: mean solar visibility, longest continuous illumination, and longest eclipse duration. Mean solar visibility is the shared solar descriptor used in both SOM formulations, whereas longest continuous illumination and longest eclipse duration are additional temporal descriptors used only in the temporally extended SOM. Mean solar visibility measures the fraction of sampled time steps during which the Sun is visible. Longest continuous illumination measures the maximum uninterrupted duration of visibility, whereas longest eclipse duration measures the maximum uninterrupted duration of darkness. As shown in Figure 1, these descriptors are assembled with the shared static lunar feature layers for subsequent SOM-based evaluation.

## 2.2. SOM-based Landing Site Evaluation

The derived visibility descriptors are incorporated into a SOM-based landing site evaluation pipeline. The baseline SOM is based on recent SOM-based south polar landing site evaluation and uses aligned engineering and scientific feature layers for unsupervised clustering and ranking analysis [10, 14, 22]. In the baseline SOM, solar visibility is represented only by mean solar visibility. In the temporally extended SOM, mean solar visibility is supplemented with two temporal descriptors, namely longest continuous illumination and longest eclipse duration, as summarized in Table 1.

Both SOM formulations use the same sliding window unit and the same static lunar feature layers, including mean Earth visibility, roughness$_{km}$, roughness$_m$, slope, iron oxide abundance, hydrogen abundance, distance to water ice, gravitational anomaly, and geological map. All layers are spatially aligned within each ROI and converted into a common sliding window representation before SOM training. This produces one feature vector for each analysis window and ensures that the baseline SOM and the temporally extended SOM are compared on the same spatial support. Feature values are normalized before training, and independent SOM models are trained for the two formulations.

The resulting node prototypes are reordered to generate ranking maps and thresholded suitability maps. After re-

ordering, lower rank values indicate more favorable landing conditions. The SOM learning mechanism itself is unchanged; only the representation of solar visibility is modified. This isolates the effect of temporal illumination information on SOM-based lunar landing site evaluation.

For coordinate-level analysis, the trained SOM outputs are further queried at specified site coordinates. Each coordinate is projected onto the same spatial grid as the SOM outputs and assigned to the corresponding analysis window. When needed, a small local neighborhood around the nearest window center is also examined to reduce sensitivity to window discretization. For both the baseline SOM and the temporally extended SOM, we extract the assigned rank, binary suitability label, and visibility descriptors at the queried location. Rank change is reported as the baseline SOM rank to the temporally extended SOM rank, and rank gain is defined as the baseline rank minus the temporally extended rank. Under our ranking convention, lower rank values indicate more favorable landing conditions; therefore, a positive gain indicates a more favorable assessment after temporal descriptors are included. This postprocessing step does not alter SOM training; it uses the same trained maps to examine how temporal descriptors affect site-level landing site assessment.

## 3. Experimental Setup

### 3.1. Study Regions and Coordinate Sets

We evaluate the proposed framework in four local regions of the lunar south polar area: Shackleton, Haworth, Cabeus, and Shoemaker. These regions were selected based on the candidate soft-landing sites proposed by Zhang *et al.* [24] and recent SOM-based evaluation efforts [14]. The regional experiments are performed using aligned local raster layers defined on a common spatial grid within each ROI.

In addition to regional map comparisons, we conduct a coordinate-level analysis using ten previously proposed landing-site coordinates from Zhang *et al.* [24] and three mission-associated coordinates: Chandrayaan-3 Statio Shiv Shakti, IM-1 near Malapert A, and IM-2 in the Mons Mouton region. The four regional ROIs are defined to include the proposed coordinates in Shackleton, Haworth, Cabeus, and Shoemaker, whereas the three mission-associated ROIs are centered on the corresponding landing or landing-attempt locations. This analysis is intended to examine how the temporally extended representation affects site-level assessment at mission-relevant locations, rather than to treat these coordinates as ground-truth labels of optimal landing sites.

Each ROI is defined as an 80 km × 80 km core area. Because lunar polar illumination depends on the local horizon, which can be affected by terrain outside the target region, temporal visibility modeling is performed with a 150 km

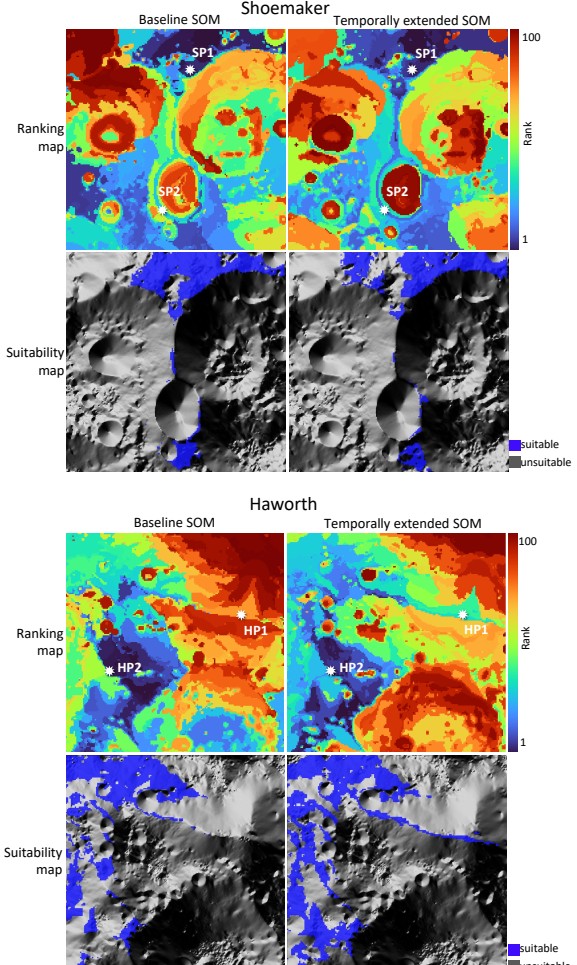

## Shoemaker

Baseline SOM     Temporally extended SOM

Ranking map

Suitability map

## Haworth

Baseline SOM     Temporally extended SOM

Ranking map

Suitability map

Figure 2. Spatial comparison of the baseline SOM and the temporally extended SOM in the Shoemaker and Haworth regions. The upper and lower rows show reordered ranking maps and thresholded suitability maps, respectively, on matched spatial extents; lower rank values indicate more favorable landing conditions. White markers denote selected proposed-site coordinates used in the coordinate-level rank comparison. The temporally extended SOM changes local rank patterns and thresholded suitable areas relative to the baseline.

surrounding buffer [8, 15, 20]. For the four proposal-based regional ROIs and the ten proposed coordinates, the temporal simulation is evaluated over one synodic cycle starting at 2020-01-01 00:00 UTC with hourly sampling. For the three mission-associated ROIs, the same synodic-cycle duration and hourly sampling are used, but the start time is set to the corresponding mission landing or landing-attempt epoch. This separates the common reference scenario used for the proposed-site regions from the mission-epoch scenarios used for the mission-associated regions.

We compare a baseline SOM and a temporally extended

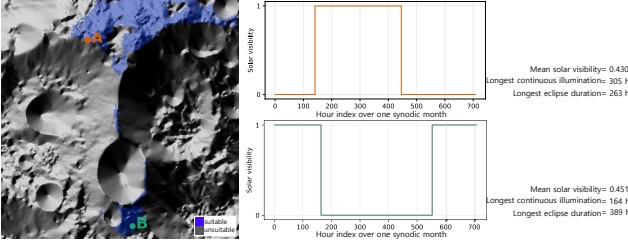

Figure 3. Temporal evidence from two representative analysis windows in the Shoemaker region. The left panel shows the selected windows on the thresholded suitability overlay, and the right panels show their binary solar visibility timelines and summary descriptors. The two windows have similar mean solar visibility but different longest continuous illumination and longest eclipse duration, illustrating that mean visibility alone can obscure operationally relevant illumination intervals.

SOM under the same sliding window unit, static feature set, and SOM-based evaluation pipeline described in Sec. 2; the only difference is the representation of solar visibility. For each formulation and each ROI, the SOM outputs are postprocessed into two map products: a ranking map and a thresholded suitability map. After SOM prototype reordering, lower rank values indicate more favorable landing conditions. To obtain binary suitability maps, we adopt the threshold screening criteria of Liu *et al*. [14]: slope $< 12°$, mean solar visibility $> 0.35$, Earth visibility $> 0.15$, and hydrogen abundance $> 100$ ppm. Because both formulations are evaluated on the same spatial support within each ROI, the resulting products can be compared directly.

### 3.2. Evaluation Protocol

Our evaluation has two components. First, we conduct a regional map-level comparison between the baseline SOM and the temporally extended SOM. We compare paired ranking maps and thresholded suitability maps in representative regions using identical spatial extents, and compare suitable-area changes across the four study regions. This comparison assesses whether temporal illumination information changes local ranking patterns and suitable area boundaries relative to the mean-visibility baseline.

Second, we conduct a coordinate-level rank comparison at ten previously proposed landing-site coordinates and three mission-associated coordinates. For each coordinate, we query the baseline SOM and temporally extended SOM ranking maps at the corresponding analysis window. Rank gain is defined as the baseline rank minus the temporally extended rank. Under our ranking convention, lower rank values indicate more favorable landing conditions; therefore, a positive gain indicates an improved rank after temporal descriptors are included. This comparison examines whether the temporal descriptors affect site-level ranking at previously proposed or mission-relevant coordinates.

Table 2. Coordinate-level rank comparison at ten previously proposed landing-site coordinates and three mission-associated coordinates. Lower rank values indicate more favorable landing conditions. Rank gain is defined as the baseline SOM rank minus the temporally extended SOM rank; positive values indicate lower, more favorable ranks after temporal descriptors are included.

| Group | ID | Baseline | Temporally extended | Rank gain |
|---|---|---|---|---|
| Mission | CY3 | 52 | 40 | +12 |
| | IM1 | 6 | 3 | +3 |
| | IM2 | 12 | 8 | +4 |
| Shackleton | P1 | 34 | 9 | +25 |
| | P2 | 18 | 15 | +3 |
| | P3 | 47 | 2 | +45 |
| | P4 | 47 | 2 | +45 |
| Cabeus | CP1 | 28 | 20 | +8 |
| | CP2 | 2 | 1 | +1 |
| Shoemaker | SP1 | 3 | 3 | 0 |
| | SP2 | 26 | 12 | +14 |
| Haworth | HP1 | 71 | 31 | +40 |
| | HP2 | 2 | 3 | −1 |
| **Rank improved / unchanged / less favorable** | | | | **11 / 1 / 1** |

# 4. Results and Discussion

Labels follow Table 2: CY3, IM1, and IM2 denote Chandrayaan-3 Statio Shiv Shakti, IM-1 near Malapert A, and IM-2 in Mons Mouton, while P, CP, SP, and HP denote Shackleton, Cabeus, Shoemaker, and Haworth coordinates.

## 4.1. Spatial Comparison in Representative Regions

Figure 2 compares the baseline SOM and the temporally extended SOM in the Shoemaker and Haworth regions. For each region, the upper row shows reordered ranking maps and the lower row shows thresholded suitability maps on the same spatial extent. Lower rank values correspond to more favorable landing conditions. The white markers indicate selected proposed-site coordinates used in the coordinate-level rank comparison.

Relative to the baseline SOM, the temporally extended SOM modifies local rank patterns in both representative regions. In Shoemaker, rank patterns around SP1 and SP2 change noticeably, while parts of the thresholded suitable area are redistributed. In Haworth, the local ranking structure around HP1 and HP2 changes together with the spatial extent of thresholded suitable areas. These results indicate that longest continuous illumination and longest eclipse duration can affect both window-level rank assignments and the spatial arrangement of suitable and unsuitable areas.

The comparison shows that temporal descriptors can change local map structure under matched spatial extents, while the corresponding coordinate-level effects are summarized in Table 2.

## 4.2. Coordinate-level Rank Comparison

Table 2 summarizes rank values sampled at ten previously proposed landing-site coordinates from Zhang *et al.* [24] and three mission-associated coordinates. The comparison uses reordered ranking maps from the baseline SOM and the temporally extended SOM. Rank change is reported as baseline rank → temporally extended rank. Because lower rank values indicate more favorable landing conditions, gain is defined as the baseline rank minus the temporally extended rank; therefore, a positive gain indicates an improved rank under the temporally extended SOM.

The temporally extended SOM assigns lower, more favorable ranks to 11 of the 13 analyzed coordinates. One coordinate remains unchanged, and one receives a slightly higher, less favorable rank. The three mission-associated coordinates all receive lower ranks under the temporally extended SOM. This suggests that temporal descriptors can affect not only regional map patterns but also site-level rank assignments at mission-relevant locations.

This comparison should not be interpreted as direct validation of landing-site optimality, because landing-site selection and mission targeting depend on mission-specific constraints that are not fully represented in the present feature set. Instead, the coordinate-level result provides a site-level case study showing that interval-based illumination descriptors can change the relative assessment of previously proposed and mission-associated coordinates.

## 4.3. Temporal Descriptor Evidence

To support the interpretation of the rank and suitability changes, we examine two representative analysis windows in the Shoemaker region with similar mean solar visibility but different illumination interval structures. This analysis illustrates why adding temporal descriptors to the SOM input representation can affect rank and suitability outcomes, even when mean solar visibility values are comparable.

As shown in Figure 3, the two selected windows have similar mean solar visibility values, but their binary visibility timelines differ substantially. Window A has a mean solar visibility of 0.430, a longest continuous illumination of 305 h, and a longest eclipse duration of 263 h. Window B has a comparable mean solar visibility of 0.451, but a shorter longest continuous illumination of 164 h and a longer longest eclipse duration of 389 h. This contrast indicates that mean solar visibility alone can obscure differences in illumination continuity and eclipse duration. When such interval-based descriptors are incorporated into the temporally extended SOM, they can alter the relative position of analysis windows in the feature space and consequently affect local rank and thresholded suitability outcomes.

## 5. Conclusion

We construct a spatiotemporal solar visibility representation over 3D terrain and incorporated mean solar visibility, longest continuous illumination, and longest eclipse duration into a SOM-based landing-site evaluation framework. Compared with the mean-visibility baseline, the temporally extended SOM changed local ranking patterns and thresholded suitability outcomes across lunar south polar regions. At ten previously proposed and three mission-associated coordinates, most coordinates received lower, more favorable ranks after temporal descriptors were included. These results suggest that mean solar visibility alone may not fully capture operationally relevant illumination structure. While the coordinate-level analysis should not be interpreted as direct validation of landing-site optimality, it shows that illumination interval information can provide additional context for regional mapping and site-level assessment.

## Acknowledgements

This work was supported by the Science Research Program using Danuri's science payloads (No. 2026185100) funded by the Korea Astronomy and Space Science Institute (KASI).

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
