# OpenReview forum: "Spatiotemporal Solar Visibility Modeling for Lunar Landing Site Evaluation"
_thecvf.com/CVPR/2026/Workshop/3D4S — CVPR 2026 Workshop 3D4S Poster_

### Official Review · Reviewer_Yuwr · 2026-04-19
**Interesting spatiotemporal solar visibility analysis with limited methodological novelty**

**Rating:** 6
**Confidence:** 3

**Review:**

## Summary

This paper presents a spatiotemporal solar visibility modeling framework for lunar landing site evaluation. The approach combines digital elevation models (DEM), terrain-aware horizon analysis, and SPICE-based solar geometry to estimate binary solar visibility over time. From this temporal representation, the authors derive visibility-related descriptors, including mean solar visibility, longest continuous illumination, and longest eclipse duration, which are further integrated into a self-organizing map (SOM) framework for ranking and suitability analysis of candidate landing sites.

The paper is motivated by the limitation of using only mean solar visibility in prior work and argues that temporal illumination structure provides additional information for decision-making in lunar polar regions.

---

## Strengths

1. **Relevant and meaningful problem setting.**
   The paper addresses an important limitation in lunar landing site evaluation, namely that mean solar visibility alone may not adequately capture operationally relevant temporal illumination characteristics.

2. **Clear and well-structured pipeline.**
   The overall workflow is logically organized, combining terrain modeling, temporal solar visibility estimation, and downstream evaluation in a coherent manner.

3. **Insightful use of temporal descriptors.**
   The introduction of interval-based descriptors (e.g., longest illumination and longest eclipse duration) provides an intuitive and interpretable extension beyond average visibility.

4. **Reasonable experimental setup.**
   The evaluation is conducted across multiple south polar regions, and qualitative as well as quantitative comparisons illustrate that temporal modeling can influence ranking and suitability outcomes.

---

## Weaknesses

1. **Limited methodological novelty.**
   The proposed approach mainly combines existing components (DEM-based horizon analysis, SPICE solar modeling, and SOM clustering) with simple temporal statistics. The contribution is largely an application-driven extension rather than a new modeling or learning framework.

2. **Lack of learning-based or optimization-based formulation.**
   The method does not introduce new representation learning, inference mechanisms, or optimization strategies, which limits its relevance to the broader computer vision community.

3. **Evaluation focuses on descriptive changes rather than clear performance gains.**
   While the results show that temporal descriptors affect ranking and suitability maps, it is less clear whether these changes lead to objectively better or more reliable outcomes.

4. **Limited generalizability beyond the specific domain.**
   The method is tailored to lunar landing site evaluation and may be difficult to transfer to other vision problems, which further limits its impact in a general CVPR context.

---

## Clarity

The paper is generally well written and easy to follow. The methodology is clearly described, and figures effectively illustrate the pipeline and results. The motivation and assumptions are also clearly stated.

---

## Overall Assessment

Overall, the paper presents a reasonable and well-executed spatiotemporal analysis framework for solar visibility modeling. While the methodological contribution is limited and primarily application-driven, the work highlights an important limitation of existing representations and provides useful insights into temporal illumination modeling.

Given its clarity, completeness, and potential to stimulate discussion in a specialized context, I believe the work is suitable for a workshop venue.

---

## Recommendation

**6: Marginally above acceptance threshold**

---

### Official Review · Reviewer_H3Q7 · 2026-04-24
**This paper aligns well with the seminar's theme and offers a valuable insight: relying solely on the metric of average solar visibility may overlook critical risks, such as prolonged periods of solar eclipse. However, the methodology employed in the paper primarily involves integrating existing tools rather than proposing an entirely novel algorithm. The authors should further clarify how temporal characteristics influence the final suitability distribution map, and provide the basic parameters for the Self-Organizing Map (SOM) or relevant results from stability tests.**

**Rating:** 7
**Confidence:** 5

**Review:**

## Review Comments

### 1. The topic aligns closely with the scope of this workshop.

This paper addresses a practical scientific 3D/4D problem: specifically, how lunar topography and time-varying solar illumination conditions influence the selection of landing sites. This aligns well with the central theme of this application-oriented 3D4S workshop.

---

### 2. The core concept of the paper is clear and has practical value.

The paper makes an insightful point: relying solely on the metric of **average solar visibility** may obscure significant potential risks, such as **prolonged solar occultation**. The comparison between two landing sites with similar average visibility but very different temporal illumination patterns is highly persuasive.

---

### 3. The methodology is practical, but somewhat limited in innovation.

The work mainly integrates **DEM-based horizon analysis**, **SPICE-derived solar geometry calculations**, **temporal feature descriptors**, and an existing **SOM framework**. This level of contribution is appropriate for a workshop paper, but the authors should avoid overstating its novelty as an entirely new method for 3D geometric analysis or a novel machine learning algorithm.

---

### 4. The suitability-map generation process needs further clarification.

The paper states that the final thresholding process relies primarily on metrics such as **slope**, **average solar visibility**, **Earth visibility**, and **hydrogen abundance**. However, the two newly introduced temporal features—**maximum continuous solar illumination duration** and **maximum continuous solar occultation duration**—do not appear to be directly incorporated into these decision rules.

The authors should explain more clearly how these two temporal features influence, and ultimately determine, whether landing areas are classified as **suitable** or **unsuitable**.

---

### 5. A simple stability test would improve the paper.

Since the **baseline SOM model** and the **temporally enhanced SOM model** are trained independently, their clustering results may be affected by SOM initialization. The authors should include a simple **multiple-random-seed stability test**. Alternatively, they should at least provide the specific SOM configuration parameters and random seed values used in the study.

---

### Official Review · Reviewer_bq6i · 2026-04-25
**The paper studies lunar landing site evaluation by modeling **spatiotemporal solar visibility** instead of relying only on mean illumination. It computes time-varying visibility using DEM-based horizon analysis and SPICE solar geometry, extracts a few temporal descriptors (e.g., longest illumination and eclipse duration), and feeds them into a SOM-based evaluation pipeline. The main takeaway is that incorporating temporal structure can change ranking and suitability outcomes compared to a mean-only baseline**

**Rating:** 6
**Confidence:** 3

**Review:**

### **Pros**

- **The problem is well motivated.**
  It makes sense that mean solar visibility alone is insufficient, and the paper clearly explains why temporal patterns (e.g., long eclipses) matter for real missions.

- **The pipeline is clean and easy to follow.**
  The comparison is well controlled (same setup, only changing solar visibility representation), which makes the experimental design straightforward and fair.


### **Cons**

- **The contribution feels incremental.**
  At the end of the day, the method mainly adds a couple of handcrafted temporal features. There is no new modeling or learning component, which limits novelty.

- **Evaluation is not very convincing.**
  The results mostly show that outputs change, but it is unclear whether they actually improve anything. There are no statistical tests or strong quantitative metrics.

- **Lack of ground truth or real validation.**
  The paper does not evaluate against any external benchmark or mission-level objective, so it’s hard to judge the practical impact of the method.

---

### **Overall Assessment**

Overall, the paper is clearly written and addresses a reasonable problem, but the contribution is fairly incremental and the evaluation is not strong enough to support a solid claim of improvement.

---

### Decision · Program_Chairs · 2026-04-28

Accept (Poster)